# Polynomial time guarantees
# for the Burer-Monteiro method

**Diego Cifuentes**
School of Industrial and Systems Engineering
Georgia Institute of Technology
Atlanta, GA 30332
diego.cifuentes@isye.gatech.edu

**Ankur Moitra**
Department of Mathematics
Massachusetts Institute of Technology
Cambridge, MA 02139
moitra@mit.edu

## Abstract

The Burer-Monteiro method is one of the most widely used techniques for solving large-scale semidefinite programs (SDP). The basic idea is to solve a nonconvex program in $Y$, where $Y$ is an $n \times p$ matrix such that $X = YY^T$. We show that this method can solve SDPs in polynomial time in a smoothed analysis setting. More precisely, we consider an SDP whose domain satisfies some compactness and smoothness assumptions, and slightly perturb the cost matrix and the constraints. We show that if $p \gtrsim \sqrt{2(1+\eta)m}$, where $m$ is the number of constraints and $\eta > 0$ is any fixed constant, then the Burer-Monteiro method can solve SDPs to any desired accuracy in polynomial time, in the setting of smooth analysis. The bound on $p$ approaches the celebrated Barvinok-Pataki bound in the limit as $\eta$ goes to zero, beneath which it the nonconvex program can be suboptimal. Our main technical contribution, which is key for our tight bound on $p$, is to connect spurious approximately critical points of the nonconvex program to tubular neighborhoods of certain algebraic varieties, and then estimate the volume of such tubes.

## 1 Introduction

Let $\mathbb{S}^n$ be the set of $n \times n$ symmetric matrices. Given matrices $A_1, \ldots, A_m, C \in \mathbb{S}^n$ and a vector $b \in \mathbb{R}^m$, consider the *semidefinite program* (SDP):

$$\min_{X \in \mathbb{S}^n} C \bullet X \quad \text{s.t.} \quad \mathcal{A}(X) = b, \quad X \succeq 0, \qquad\qquad (SDP)$$

where $\mathcal{A} : \mathbb{S}^n \to \mathbb{R}^m$, $X \mapsto (A_1 \bullet X, \ldots, A_m \bullet X)$ is a linear map and the notation $X \succeq 0$ in indicates that $X$ is a positive semidefinite (PSD) matrix. Also consider the *least squares* SDP

$$\min_{X \in \mathbb{S}^n} \|\mathcal{A}(X) - b\|^2 \quad \text{s.t.} \quad X \succeq 0, \qquad\qquad (SDP_{ls})$$

Though interior point methods solve (SDP) and ($SDP_{ls}$) in polynomial time, they often have memory problems for large values of $n$. The Burer-Monteiro method [11, 12] is one of the most widely used procedures for large scale problems. Several papers have worked in understanding the practical success of this method, e.g., [9, 10, 12]. Although various results have been shown, they all fall short of showing that one can reach an approximately optimal solution of (SDP)/($SDP_{ls}$) in polynomial time. This paper provides the first polynomial time guarantees for the Burer-Monteiro method.

The Burer-Monteiro method consists in writing $X = YY^T$ for some $Y \in \mathbb{R}^{n \times p}$, and solving a nonconvex optimization problem in terms of $Y$. For (SDP) we get the problem

$$\min_{Y \in \mathfrak{M}} C \bullet YY^T, \qquad \mathfrak{M} := \{Y \in \mathbb{R}^{n \times p} : \mathcal{A}(YY^T) = b\}, \qquad\qquad (BM)$$

and for ($SDP_{ls}$) we get the unconstrained problem

$$\min_{Y \in \mathbb{R}^{n \times p}} \|\mathcal{A}(YY^T) - b\|^2. \qquad (BM_{ls})$$

We focus now on problem (*SDP*), though our results apply to ($SDP_{ls}$) as well. Let $\tau(k) := \binom{k+1}{2}$ be the $k$-th triangular number. It is known that problems (*SDP*) and (*BM*) have the same optimal value for any $p$ such that $\tau(p) \geq m$; this is known as the *Barvinok-Pataki bound* [3, 29]. But due to nonconvexity, local optimization methods may converge to a critical point of (*BM*) which is not globally optimal [34]. In this paper we are mainly interested in 2nd-order critical points (abbreviated as 2-critical points).

Problem (*BM*) has been well studied, most remarkably in the sequel of papers [5, 9, 10, 30]. Boumal et al. [9, 10] showed that (*BM*) has no spurious 2-critical points when $\tau(p) > m$, assuming that the feasible set $\mathfrak{M}$ is a smooth manifold and that the cost matrix $C$ is *generic*. Hence, the Burer-Monteiro method converges to the global optimal solution as long as the cost $C$ is outside a certain "bad" set, which has measure zero. However, this analysis only applies to the limit points. The follow up works [5] and [30] provide finite-iteration guarantees, though for values of $p$ larger than the Barvinok-Pataki bound. They introduce a smoothed analysis [32] setting in which the cost matrix $C$ is subject to a small random perturbation of magnitude $\sigma$. Such a perturbation ensures that $C$ stays outside the "bad" region, so the limit points are globally optimal. They then argue that certain notions of approximate criticality for (*BM*) guarantee approximate optimality for (*SDP*) when $\tau(p) \gtrsim \frac{9}{2} m \log(\Omega(\sqrt{n}/\sigma))$.

Despite the earlier work, the question of whether the Burer-Monteiro method can be used to completely solve problem (*SDP*) in polynomial time has not been yet resolved. The works [5, 30] do not provide end-to-end polynomial time guarantees and make structural assumptions on the problem, as will be further discussed in the related work section. In particular, finding a feasible solution to (*SDP*) that could serve as initial point is a complicated and practically relevant problem, but it has not been addressed before. In addition, the lower bound on $p$ is qualitatively wrong, as it is not only larger than the Barvinok-Pataki bound, but also gets worse as the magnitude $\sigma$ of the perturbation decreases.

In order to fully solve (*SDP*), we propose a 2-stage procedure. First, solve ($BM_{ls}$) to either get a point $Y_0$ that is approximately feasible, i.e.,. $\mathcal{A}(Y_0 Y_0^T) \approx b$, or to prove that no such point exists. Second, solve (*BM*) to get an approximately optimal point, starting with point $Y_0$ from the first stage. We prove that this scheme is successful when $\tau(p) > (1+\eta)m$, for an arbitrary constant $\eta > 0$.

The next theorem provides polynomial time guarantees for the second stage. The complexity is with respect to the setting of a smoothed analysis, in which the cost matrix $C$ is perturbed.

**Theorem 1** ((*SDP*) in polytime)**.** *Let $p$ such that $\tau(p) > (1+\eta)m$, for a fixed $\eta > 0$. Apply a random perturbation of magnitude $\sigma$ to the cost matrix $C$. Assume that $\mathfrak{M}$ is compact and smooth (LICQ holds). Solve (BM) using a constrained optimization method with 2nd-order guarantees (e.g., Theorem 4) initialized at an approximately feasible point $Y_0$. Then, after $\mathrm{poly}(n, \sigma^{-1})$ iterations, we get a point $Y$ such that $YY^T$ is approximately optimal for (SDP) with high probability.* A more detailed version of this result, with explicit optimality and complexity constants, appears in Theorem 9.

Note that our bound on $p$ does not depend on the magnitude $\sigma$ of the perturbation, as opposed to earlier works. This improved bound relies on a technical novelty of this paper, namely, using sophisticated tools from real algebraic geometry for concentration bounds. Indeed, we characterize the presence of spurious approximately critical points through tubular neighborhoods of algebraic varieties, and make use of effective bounds for the volume of such tubes [4, 23, 27]. On the other hand, [9, 10] uses simple dimension arguments, while [5, 30] rely on random matrix theory.

The following theorem provides polynomial time guarantees for the first stage. We again consider a smoothed analysis setting, but this time perturbing the matrices $A_i$ that define the linear map $\mathcal{A}$.

**Theorem 2** (($SDP_{ls}$) in polytime)**.** *Let $p$ such that $\tau(p) > (1+\eta)m$, for a fixed $\eta > 0$. Apply a random perturbation of magnitude $\sigma$ to the constraint map $\mathcal{A}$. Assume that the sublevel sets $\{Y : \|\mathcal{A}(YY^T) - b\| \leq \alpha\}$ are compact. Solve ($BM_{ls}$) using an unconstrained optimization method with 2nd-order guarantees (e.g., [15, 16]). Then, after $\mathrm{poly}(n, \sigma^{-1})$ iterations, we get a point $Y$ such that $YY^T$ is approximately optimal for ($SDP_{ls}$) with high probability.* A more detailed version of this result, with explicit optimality and complexity constants, appears in Theorem 10.

Theorems 1 and 2 together provide end-to-end guarantees for solving (*SDP*), with polynomial smoothed analysis complexity. The main assumptions are that $p$ is slightly above the Barvinok-Pataki bound, while the feasible set is compact and smooth. In the general case, we require that both $C, \mathcal{A}$ are slightly perturbed. If a feasible point of the (*SDP*) is known, then Theorem 1 alone is enough.

We point out that problem ($SDP_{ls}$) is a matrix sensing problem which is of interest in its own right, independent of its connection to SDP feasibility. The guarantees from Theorem 2 apply even if the optimal value of ($SDP_{ls}$) is strictly positive (i.e., there is no $X \succeq 0$ with $\mathcal{A}(X)=b$). There are earlier works [6, 22, 26] proving global optimality guarantees for the nonconvex problem ($BM_{ls}$), but they all rely on the restricted isometry property (RIP). To the best of our knowledge, Theorem 2 provides the first global guarantees for ($BM_{ls}$) that do not rely on RIP.

**Related work**

We first discuss (*SDP*). Early work by Burer and Monteiro [12] and by Journée et al. [25] gave strong indications that (*BM*) has no spurious critical points above the Barvinok-Pataki bound. A formal proof was given by Boumal et al. [5, 9] under a genericity assumption. Subsequent work by Bhojanapalli et al. [5] and Pumir et al. [30] investigated the case of approximately critical points under smoothed analysis. The Barvinok-Pataki bound was recently shown to be optimal up to lower order terms for general SDPs [34], though for structured families of SDPs a smaller rank might suffice [2, 28]. The paper [19] extended the results from [9] to SDPs with multiple PSD constraints.

The works [5, 30] are the closest to this paper, but they do not provide end-to-end polynomial guarantees for solving general SDPs (nor it is claimed). In particular, neither of them discuss initialization or feasibility. The paper [30] relies on Riemannaian optimization, which is only practical for very special SDPs (like the maxcut SDP). In the general case, the retraction operator is computationally intractable. The paper [5] relies on a penalized version of (*SDP*). It has the crucial catch that they require a preprocessing step that in general is as hard as solving another SDP. Since solving an auxiliary SDP is required, it fails to explain why the BM method improves over general SDP solvers. In addition, the optimality guarantees in [5] are with respect to the solution of the penalized problem. Translating their results into polynomial time guarantees with respect to the original SDP is nontrivial due to interdependence between the penalty parameter, the perturbation magnitude, and the target precision[1]. In contrast, our paper does provide end-to-end polynomial time guarantees, and even better, our results hold down to the Barvinok-Pataki bound.

Problem ($SDP_{ls}$) has been well studied in the matrix sensing community. Bhojanapalli et al. [6] showed that ($BM_{ls}$) has no spurious local minima under the RIP condition, and they also provided polynomial time guarantees. Similar results have been derived later, e.g., [22, 26]. Note that RIP is a very strong assumption about the condition number of the linear map $\mathcal{A}$, particularly since the RIP constant needs to be small [35]. In contrast, our result in Theorem 2 simply assumes a small perturbation around a worst-case instance $\mathcal{A}$. The comparison of smoothed analysis with RIP is akin to assuming that a matrix has a nonzero smallest singular value (and getting bounds that depend on inverse polynomials on this quantity), rather than assuming its condition number is close to one.

## 2   Critical points in nonlinear programming

**Unconstrained case.**  Consider the optimization problem

$$\min_{y \in \mathbb{R}^n} f(y), \tag{$P_{\mathrm{un}}$}$$

with $f : \mathbb{R}^n \to \mathbb{R}$ twice continuously differentiable. A vector $y \in \mathbb{R}^n$ is a *2nd-order critical* point for ($P_{\mathrm{un}}$), abbreviated 2-critical, if $\nabla f(y) = 0$, $\nabla^2 f(y) \succeq 0$. Practical optimization algorithms cannot obtain a solution satisfying these equations exactly. Hence, we relax these conditions.

**Definition 1.** Given $\varepsilon_1, \varepsilon_2 \in \mathbb{R}_+$, a vector $y$ is $(\varepsilon_1, \varepsilon_2)$-*approximately 2-critical* (AC) for ($P_{\mathrm{un}}$) if:

$$\|\nabla f(y)\| \leq \varepsilon_1, \qquad \nabla^2 f(y) \succeq -\varepsilon_2 I_n. \tag{1}$$

---

[1]Translating optimality guarantees of the penalized problem to the original SDP requires tuning the penalty parameter $\mu$. The straightforward choice is $\mu = \Omega(\epsilon^{-2})$, where $\epsilon$ is the target precision. This leads to high Lipschitz numbers, complicating the complexity analysis (e.g., Theorem 3 does not apply). In addition, [5, Thm 13] constraints the perturbation magnitude by $\sigma^2 \geq \Omega(\mu \epsilon^{3/2} \gamma^{-1}) \geq \Omega(\epsilon^{-1/2} \gamma^{-1})$. Hence, the results seem to require large perturbations, while in smoothed analysis we are interested in small perturbations.

Several algorithms for unconstrained optimization with provable convergence guarantees are known. Recent work has focused on deriving algorithms with finite-time guarantees. In particular, the trust region method computes an $(\varepsilon_1, \varepsilon_2)$-AC point in $O(\max\{\varepsilon_1^{-2}\varepsilon_2^{-1}, \varepsilon_2^{-3}\})$ iterations [16], and the adaptive regularization with cubics (ARC) method takes $O(\max\{\varepsilon_1^{-2}, \varepsilon_2^{-3}\})$ iterations [15]. We formally state the result for the ARC method.

**Theorem 3** ( [15]). *Assume that there exists $\alpha > 0$ such that a point $y_0$ with $f(y_0) \leq \alpha$ is known, and the functions $f, \nabla f, \nabla^2 f$ are uniformly bounded and Lipschitz continuous on the set $\{y : f(y) \leq \alpha\}$. The ARC method initialized at $y_0$ requires $O(\max\{\varepsilon_1^{-2}, \varepsilon_2^{-3}\})$ iterations to produce an $(\varepsilon_1, \varepsilon_2)$-AC point $y$. Furthermore, each iteration requires $O(1)$ evaluations of $f$ and its derivatives.*

**Constrained case.** Consider the nonlinear program

$$\min_{y \in \mathfrak{M}} \; f(y), \qquad \mathfrak{M} := \{y \in \mathbb{R}^m : h(y) = 0\}, \qquad (P_{\text{con}})$$

with $f : \mathbb{R}^n \to \mathbb{R}$, $h : \mathbb{R}^n \to \mathbb{R}^m$ twice continuously differentiable. The Lagrangian function is $L(y, \lambda) = f(y) + \lambda \cdot h(y)$. A vector $y \in \mathbb{R}^n$ is a *2-critical point* if there are multipliers $\lambda \in \mathbb{R}^m$ satisfying some first order and second order optimality conditions. As before, for practical optimization we need to consider a notion of approximate criticality.

**Definition 2.** Given $\varepsilon = (\varepsilon_0, \varepsilon_1, \varepsilon_2) \in \mathbb{R}_+^3$, $\gamma \in \mathbb{R}_+$, a pair $(y, \lambda)$ is $(\varepsilon, \gamma)$-*approximately feasible approximately 2-critical* (AFAC) for $(P_{\text{con}})$ if:

$$\|h(y)\| \leq \varepsilon_0, \qquad \|\nabla_y L(y, \lambda)\| \leq \varepsilon_1, \qquad (2a)$$

$$u^T \nabla_{yy}^2 L(y, \lambda) u \geq -\varepsilon_2 \quad (\forall \, u : \|u\| = 1, \|\nabla h(y) u\| \leq \gamma). \qquad (2b)$$

In the constrained case, for a local minima to be a critical point we need some regularity conditions. One such condition is the *linear independence constraint qualification* (LICQ), that states that $\nabla h(y)$ is full rank. This is equivalent to $\mathfrak{M}$ being smooth at $y$ for the case of complete intersections (i.e., $\text{codim}\,\mathfrak{M} = m$). We next introduce a quantitative version of LICQ.

**Definition 3.** For $\varrho > 0$, say that $\varrho$-*LICQ* holds at $y$ if the singular values of $\nabla h(y)$ are at least $\varrho$.

Several local optimization methods for $(P_{\text{con}})$ with provable convergence guarantees are known. In particular, augmented Lagrangians [1] and trust-region methods [20, §15.4] converge to 2-critical points. Recent work has focused on finding algorithms with finite-time guarantees. The complexity of computing approximately 1-critical points was studied in, e.g., [7, 17, 18, 21].

As for approximately 2-critical points, we are only aware of [14, 31]. But both papers use a different 2nd-order condition, which is not easy to translate into our setting. Nonetheless, in Theorem 4 below we show that AFAC points can be computed in polynomial time. The proof of this theorem is in Appendix A, and relies on a variant of the method from [14]. To the best of our knowledge, this is the first polynomial time bound for computing 2-critical points using the standard notions of criticality.

**Theorem 4.** *Assume that there exist $\beta, \varrho \in \mathbb{R}_+$ such that a point $y_0$ in the set $\mathfrak{M}_\beta := \{y : \|h(y)\| \leq \beta\}$ is known, the functions $f, \nabla f, \nabla^2 f, \nabla h_i, \nabla^2 h_i \, (i \in [m])$ are uniformly bounded and Lipschitz continuous on $\mathfrak{M}_\beta$, and $\varrho$-LICQ holds at all $y \in \mathfrak{M}_\beta$. Let $\varepsilon = (\varepsilon_0, \varepsilon_1, \varepsilon_2) \in \mathbb{R}_+^3$, $\gamma \in \mathbb{R}_+$ such that*

$$\varepsilon_0 \leq \beta, \quad \varepsilon_1 \leq 1, \quad \varepsilon_1^2 \leq \tfrac{1}{16} R_\lambda^{-1} \varepsilon_0 \, \varepsilon_2, \quad \gamma^2 \leq \tfrac{1}{16} R_\lambda^{-3} \varepsilon_0 \, \varepsilon_2, \qquad (3)$$

*where $R_\lambda := 2 + 2\varrho^{-1} L_f$, $L_f := \max_y \|\nabla f(y)\|$. There is an algorithm that, when initialized at $y_0$, requires $O(\max\{\varepsilon_0^{-2} \varepsilon_1^{-2}, \varepsilon_0^{-3} \varepsilon_2^{-3}\})$ evaluations of $f, h$ and their derivatives to produce an $(\varepsilon, \gamma)$-AFAC pair $(y, \lambda)$, with $\|\lambda\| \leq R_\lambda$.*

*Remark.* Theorem 4 assumes that $\varrho$-LICQ holds everywhere in $\mathfrak{M}_\beta$, an assumption which can only be guaranteed when $\beta$ is very small. Hence the initial point $y_0$ must be approximately feasible.

## 3 Optimality of critical points of (*BM*)

In this section we will show that problem (*BM*) has no *spurious* approximately feasible approximately critical (AFAC) points with high probability. This means that any AFAC point of (*BM*) is approximately optimal for (*SDP*). We consider a smoothed analysis setting in which the constraint

variables $\mathcal{A}, b$ are fixed, and the cost matrix $C$ is subject to a small random perturbation. We will restrict our attention to AFAC pairs $(Y, \lambda)$ of bounded norm. Hence, we assume that $\|Y\| \leq R_Y$ and $\|\lambda\| \leq R_\lambda$ for some fixed constants $R_Y, R_\lambda$.

A crucial step toward our theorem is a geometric characterization of the spurious AFAC points in terms of tubes around algebraic varieties. We then take advantage of known effective bounds for the volume of such tubes [4, 27].

From now on, we use the Frobenius norm for all matrices.

## 3.1 Spurious approximately critical points

The optimality conditions for (*SDP*) are well known: $\mathcal{A}(X) = b, S(\lambda)X = 0, X \succeq 0, S(\lambda) \succeq 0$, where $S(\lambda)$ is the following *slack* matrix

$$S(\lambda) := C - \mathcal{A}^*(\lambda) \in \mathbb{S}^n,$$

and $\mathcal{A}^* : \mathbb{R}^m \to \mathbb{S}^n, \lambda \mapsto \sum_i \lambda_i A_i$ is the adjoint of $\mathcal{A}$. We now relax the optimality conditions.

**Definition 4.** Let $\boldsymbol{\varepsilon} = (\varepsilon_0, \varepsilon_1, \varepsilon_2) \in \mathbb{R}^3_+$. A pair $(X, \lambda)$ is $\boldsymbol{\varepsilon}$-*approximately optimal* for (*SDP*) if:

$$\|\mathcal{A}(X) - b\| \leq \varepsilon_0, \quad \|S(\lambda)X\| \leq \varepsilon_1, \quad X \succeq 0, \quad S(\lambda) \succeq -\varepsilon_2 I_n. \tag{4}$$

It is known that an $\boldsymbol{\varepsilon}$-approximately optimal solution is at distance $O(\|\boldsymbol{\varepsilon}\|)$ from an optimal solution under nondegeneracy assumptions [33]. We can also give a simple bound on the optimality gap.

**Lemma 1.** *If $(\bar{X}, \bar{\lambda})$ is $\boldsymbol{\varepsilon}$-approximately optimal for* (*SDP*) *then*

$$C \bullet \bar{X} \leq C \bullet X + \varepsilon_0 \|\bar{\lambda}\| + \varepsilon_1 \sqrt{n} + \varepsilon_2 \|X\| \sqrt{n} \qquad \forall \text{ feasible } X.$$

*Proof.* The lemma follows from the following equations:

$$
\begin{aligned}
C \bullet X &= \bar{\lambda} \cdot \mathcal{A}(X) + S(\bar{\lambda}) \bullet X \geq \bar{\lambda} \cdot b - (\varepsilon_2 I_n) \bullet X = \bar{\lambda} \cdot b - \varepsilon_2 \|X\| \sqrt{n}, \\
\bar{\lambda} \cdot b &\geq \bar{\lambda} \cdot \mathcal{A}(\bar{X}) - \|\bar{\lambda}\| \, \|b - \mathcal{A}(\bar{X})\| \geq \bar{\lambda} \cdot \mathcal{A}(\bar{X}) - \varepsilon_0 \|\bar{\lambda}\|, \\
\bar{\lambda} \cdot \mathcal{A}(\bar{X}) &\geq \bar{\lambda} \cdot \mathcal{A}(\bar{X}) + S(\bar{\lambda}) \bullet \bar{X} - \|S(\bar{\lambda})\bar{X}\|_* \geq C \bullet \bar{X} - \varepsilon_1 \sqrt{n}. \qquad \square
\end{aligned}
$$

We proceed to problem (*BM*). This is a special instance of the nonlinear program ($P_{\text{con}}$) with $f(Y) = C \bullet YY^T$ and $h(Y) = \mathcal{A}(YY^T) - b$. The Lagrangian function is $L(y, \lambda) = S(\lambda) \bullet YY^T + b^T \lambda$. The criticality conditions for (*BM*) are obtained by specializing (2).

**Definition 5.** Let $\boldsymbol{\varepsilon} = (\varepsilon_0, \varepsilon_1, \varepsilon_2) \in \mathbb{R}^3_+, \gamma \in \mathbb{R}_+$. A pair $(Y, \lambda)$ is $(\boldsymbol{\varepsilon}, \gamma)$-AFAC for (*BM*) if:

$$\|\mathcal{A}(YY^T) - b\| \leq \varepsilon_0, \qquad \|S(\lambda)Y\| \leq \varepsilon_1, \tag{5a}$$

$$S(\lambda) \bullet UU^T \geq -\varepsilon_2 \qquad (\forall U \in \mathbb{R}^{n \times p} : \|U\| = 1, \|\mathcal{A}(UY^T)\| \leq \gamma). \tag{5b}$$

We are ready to formalize the concept of spurious critical points.

**Definition 6.** Let $R_Y, R_\lambda \in \mathbb{R}_+$ be fixed and let $(Y, \lambda)$ such that $\|Y\| \leq R_Y$ and $\|\lambda\| \leq R_\lambda$. Given $\boldsymbol{\varepsilon} = (\varepsilon_0, \varepsilon_1, \varepsilon_2) \in \mathbb{R}^3_+, \gamma \in \mathbb{R}_+$, $(Y, \lambda)$ is *spurious* $(\boldsymbol{\varepsilon}, \gamma)$-AFAC if $(Y, \lambda)$ is $(\boldsymbol{\varepsilon}, \gamma)$-AFAC for (*BM*) but $(YY^T, \lambda)$ is not $\boldsymbol{\varepsilon}'$-approx. optimal for (*SDP*) for $\boldsymbol{\varepsilon}' := (\varepsilon_0, R_Y \varepsilon_1, \varepsilon_2)$. A pair $(Y, \lambda)$ is spurious *exactly* critical if the above holds for $\boldsymbol{\varepsilon} = 0, \gamma = 0$.

## 3.2 Statement of the theorem

We present the main result of this section. Let $\mathcal{A}, b$ be fixed, and let $C$ be obtained from a random perturbation of magnitude $\sigma$ around some fixed $\bar{C}$. Concretely, $C$ is *uniformly* distributed on the Frobenius ball $\mathbf{B}_\sigma(\bar{C}) \subset \mathbb{S}^n$ of radius $\sigma$ centered at $\bar{C}$. Consider the set $\mathscr{C}_{\boldsymbol{\varepsilon}, \gamma} \subset \mathbb{S}^n$, consisting of all cost matrices for which there is a spurious AFAC point:

$$\mathscr{C}_{\boldsymbol{\varepsilon}, \gamma} := \{C \in \mathbb{S}^n : \exists (Y, \lambda) \text{ an spurious } (\boldsymbol{\varepsilon}, \gamma)\text{-AFAC pair}\}.$$

We show that if $\tau(p) > m$, then the probability $\Pr[C \in \mathscr{C}_{\boldsymbol{\varepsilon}, \gamma}] \to 0$ as the ratio $\varepsilon_1/\gamma \to 0$.

**Theorem 5** (critical ⇒ optimal). *Let $p$ such that $\tau(p) > m$. Let $\varepsilon \in \mathbb{R}_+^3$, $\gamma \in \mathbb{R}_+$. Let $C$ be uniformly distributed on the Frobenius ball $\mathbf{B}_\sigma(\bar{C})$. Then*

$$\Pr[C \in \mathscr{C}_{\boldsymbol{\varepsilon},\gamma}] \; \leq \; 4e\,\delta^{\tau(p)-m}\,(3\kappa)^m\,(4n^3/\sigma)^{\tau(p)},$$

*where $\delta := \varepsilon_1 \|\mathcal{A}\|/\gamma$ and $\kappa := R_\lambda \|\mathcal{A}\|$, provided that $\delta < \sigma/4n^3$.*

The following corollary shows that when the stronger condition $\tau(p) > (1+\eta)m$ holds, where $\eta$ is a fixed constant, then we can derive a high probability bound while maintaining $\delta$ polynomially bounded. Its proof is a straightforward manipulation.

**Corollary 1.** *Consider the setup from Theorem 5. Assume that $\tau(p) \geq (1+\eta)m + \eta t$ and $\delta \leq (1/3\kappa)^{1/\eta}(\sigma/4n^3)^{1+1/\eta}$ for some $\eta, t > 0$. Then $\Pr[C \in \mathscr{C}_{\boldsymbol{\varepsilon},\gamma}] \; \leq \; 4e\,(\sigma/12\kappa n^3)^t$.*

### 3.3 Tubes around varieties

Our proof of Theorem 5 relies on a geometric characterization of the set $\mathscr{C}_{\boldsymbol{\varepsilon},\gamma}$. Such characterization is known for the case $\varepsilon = 0, \gamma = 0$, corresponding to *exactly* critical points. It was shown in [10], see also [19], that the existence of a spurious exactly critical point implies that $C$ lies in a certain algebraic variety of $\mathbb{S}^n$, as follows:

$$\exists\ (\text{spurious exactly critical point}) \quad \Longrightarrow \quad C \in \mathbb{S}_{n-p}^n + \operatorname{Im}\mathcal{A}^*,$$

where $\mathbb{S}_{n-p}^n := \{X : \operatorname{rank} X \leq n-p\}$ is a variety of bounded rank matrices, and $\operatorname{Im}\mathcal{A}^*$ is the linear space spanned by $A_1, \ldots, A_m$. Hence, we have that $\mathscr{C}_{0,0} \subset \mathbb{S}_{n-p}^n + \operatorname{Im}\mathcal{A}^*$. When $\tau(p) > m$, the variety $\mathbb{S}_{n-p}^n + \operatorname{Im}\mathcal{A}^*$ is properly contained in $\mathbb{S}^n$. It follows that $\mathscr{C}_{0,0}$ has measure zero and hence, for generic $C$, there are no spurious exactly critical points.

We show below that for approximately critical points the situation is analogous, except that we need to consider a *tubular* neighborhood around the variety $\mathbb{S}_{n-p}^n + \operatorname{Im}\mathcal{A}^*$.

**Definition 7.** Given $\mathcal{W} \subset \mathbb{S}^n$, $\delta \in \mathbb{R}_+$, let $\operatorname{tube}_\delta \mathcal{W} := \{X \in \mathbb{S}^n : \exists W \in \mathcal{W} \text{ s.t. } \|X - W\| \leq \delta\}$.

**Proposition 1.** *Let $\delta := \varepsilon_1 \|\mathcal{A}\|/\gamma$ and $B_\lambda := \{\lambda \in \mathbb{R}^m : \|\lambda\| \leq R_\lambda\}$. Then*

$$\mathscr{C}_{\boldsymbol{\varepsilon},\gamma} \subset \operatorname{tube}_\delta(\mathbb{S}_{n-p}^n) + \mathcal{A}^*(B_\lambda).$$

The next lemma is the key ingredient for Proposition 1. It shows that if $Y$ is a spurious AFAC point, then its smallest singular value $\sigma_p(Y)$ is bounded from below. This is a generalization of a previous result by Burer and Monteiro [12] about exactly critical points. An analogue of this lemma is found in [30, Lem.3.2] for the case where $\varepsilon_0 = 0$.

**Lemma 2.** *Let $Y$ be an $(\varepsilon, \gamma)$-AFAC point of (BM). If $\sigma_p(Y) \leq \gamma/\|\mathcal{A}\|$, then $YY^T$ is $\varepsilon'$-approximately optimal for (SDP), with $\varepsilon' := (\varepsilon_0, R_Y \varepsilon_1, \varepsilon_2)$.*

*Proof.* Let $(Y, \lambda)$ satisfy (5), and let us show that $(YY^T, \lambda)$ satisfies (4). The first three conditions in (4) are easy to check. We proceed to show the last one: $S(\lambda) \succeq -\varepsilon_2 I_n$. Given a unit vector $u \in \mathbb{R}^n$, we need to show that $u^T S(\lambda) u \geq -\varepsilon_2$. There is a unit vector $z \in \mathbb{R}^p$ such that $\|Yz\| = \sigma_p(Y)$. The matrix $U := uz^T$ satisfies $\|U\| = 1$ and

$$\|UY^T\| \leq \|u\|\|Yz\| = \sigma_p(Y) \leq \gamma/\|\mathcal{A}\|.$$

Then $\|\mathcal{A}(UY^T)\| \leq \gamma$, so by (5b) we have

$$-\varepsilon_2 \; \leq \; S(\lambda) \bullet UU^T \; = \; \|z\|^2 (u^T S(\lambda) u) \; = \; u^T S(\lambda) u. \qquad \square$$

*Proof of Proposition 1.* Let $C \in \mathscr{C}_{\boldsymbol{\varepsilon},\gamma}$, so there is a spurious $(\varepsilon, \gamma)$-AFAC pair $(Y, \lambda)$. By Lemma 2, we must have $\sigma_p(Y) > \gamma/\|\mathcal{A}\|$. Let $S := S(\lambda)$, and note that $\|SY\| \leq \varepsilon_1 = \gamma \delta/\|\mathcal{A}\|$ by (5a). Then

$$\operatorname{dist}(S, \mathbb{S}_{n-p}^n) \; = \; \sqrt{\textstyle\sum_{i=1}^p \sigma_{n-i+1}^2(S)} \; \leq \; \|SY\|/\sigma_p(Y) \; < \; \delta, \tag{6}$$

so that $C = S(\lambda) + \mathcal{A}^*(\lambda) \in \operatorname{tube}_\delta(\mathbb{S}_{n-p}^n) + \mathcal{A}^*(B_\lambda)$. $\qquad \square$

By Proposition 1, the set of "bad" cost matrices $\mathscr{C}_{\boldsymbol{\varepsilon},\gamma}$ is contained in a tube around a variety. To prove Theorem 5, we need an upper bound on the probability mass of this tube. The computation of integrals over tubes has a long history in differential geometry [23]. Effective bounds for these integrals were shown in [4, 13, 27]; they were used for smooth analysis in the first reference. We will use the following recent bound.

**Theorem 6** ( [4, Thm.1.1]). *Let $V \subset \mathbb{R}^k$ be a real variety of codimension $c$ defined by polynomials of degree at most $D$. Let $x$ be uniformly distributed on the Euclidean ball $\mathbf{B}_\sigma(\bar{x}) \subset \mathbb{R}^k$. If the ratio $\sigma/\delta \geq (4D+1)(k-c)$, then we have that $\Pr[x \in \mathrm{tube}_\delta(V)] \leq 4e(4kD\delta/\sigma)^c$,*

*Proof of Theorem 5.* Let $B_\lambda \subset \mathbb{R}^m$ be the ball of radius $R_\lambda$ centered at zero. Consider an $\epsilon$-net $\mathcal{N}$ of $B_\lambda$, where $\epsilon := \delta/\|\mathcal{A}\|$. It is known that $(3R_\lambda/\epsilon)^m = (3\kappa/\delta)^m$ points suffices. Observe that

$$\mathcal{A}^*(B_\lambda) \subset \mathcal{A}^*(\mathrm{tube}_\epsilon(\mathcal{N})) \subset \mathrm{tube}_\delta(\mathcal{A}^*(\mathcal{N})), \tag{7a}$$

$$\mathscr{C}_{\boldsymbol{\varepsilon},\gamma} \subset \mathrm{tube}_\delta(\mathbb{S}^n_{n-p}) + \mathcal{A}^*(B_\lambda) \subset \mathrm{tube}_{2\delta}(\mathbb{S}^n_{n-p}) + \mathcal{A}^*(\mathcal{N}). \tag{7b}$$

Recall that $\mathbb{S}^n_{n-p}$ is a variety of codimension $\tau(p)$ defined by equations of degree $n-p+1$. For any $\ell \in \mathcal{N}$, Theorem 6 gives

$$\Pr[C - \mathcal{A}^*(\ell) \in \mathrm{tube}_{2\delta}(\mathbb{S}^n_{n-p})] \; < \; 4e\big(8\tau(n)(n-p+1)\delta/\sigma\big)^{\tau(p)} \; < \; 4e\big(4n^3\delta/\sigma\big)^{\tau(p)}.$$

Finally, the union bound gives

$$\Pr[C \in \mathscr{C}_{\boldsymbol{\varepsilon},\gamma}] \; \leq \; \#\mathcal{N} \cdot \Pr[C \in \mathrm{tube}_{2\delta}(\mathbb{S}^n_{n-p}) + \mathcal{A}^*(\ell)] \; < \; (3\kappa/\delta)^m \cdot 4e\left(4n^3\delta/\sigma\right)^{\tau(p)}. \qquad \square$$

# 4 Optimality of critical points of $(BM_{ls})$

In this section we will show that $(BM_{ls})$ has no *spurious* approximately critical (AC) points with high probability. This means that any AC point of $(BM_{ls})$ is approximately optimal for $(SDP_{ls})$. Hence, we can solve the least squares problem $(SDP_{ls})$ to global optimality using the Burer-Monteiro method. Note that if $(SDP)$ is feasible, then the optimal value of $(SDP_{ls})$ is zero. Nevertheless, the results of this section apply to an arbitrary instance of $(SDP_{ls})$, even if the optimal value is nonzero. Appendix B contains the proofs of the lemmas, propositions, and theorems from this section.

We proceed to formalize the notion of spurious critical points. The optimality conditions for the convex problem $(SDP_{ls})$ are: $S(X)X = 0, X \succeq 0, S(X) \succeq 0$, where $S(X)$ is the gradient of the least squares objective function $f(X) = \|\mathcal{A}(X) - b\|^2$:

$$S(X) := \nabla f(X) = 2\mathcal{A}^*(\mathcal{A}(X) - b) \in \mathbb{S}^n.$$

We call $X$ approximately optimal if either $\mathcal{A}(X) \approx b$ or the above conditions are almost satisfied.

**Definition 8.** Let $\boldsymbol{\varepsilon} = (\varepsilon_0, \varepsilon_1, \varepsilon_2) \in \mathbb{R}^3_+$. A matrix $X \succeq 0$ is $\boldsymbol{\varepsilon}$-*approximately optimal* for $(SDP_{ls})$ if

$$\|\mathcal{A}(X) - b\| \leq \varepsilon_0 \qquad \text{or} \qquad (\; \|S(X)X\| \leq \varepsilon_1 \; \text{and} \; S(X) \succeq -\varepsilon_2 I_n \;). \tag{8}$$

The following lemma bounds the optimality gap for the second case in (8).

**Lemma 3.** *Let $\bar{X} \in \mathbb{S}^n_+$ such that $\|S(\bar{X})\bar{X}\| \leq \varepsilon_1$, $S(\bar{X}) \succeq -\varepsilon_2 I_n$. Then*

$$f(\bar{X}) \; \leq \; f(X) + \varepsilon_1\sqrt{n} + \varepsilon_2\|X\|\sqrt{n} \qquad \forall\, X \in \mathbb{S}^n_+.$$

The Burer-Monteiro problem $(BM_{ls})$ is a special instance of the unconstrained optimization problem $(P_{\mathrm{un}})$. The criticality conditions for $(BM_{ls})$ are obtained by specializing (1).

**Definition 9.** *Let $(\varepsilon_1, \varepsilon_2) \in \mathbb{R}^2_+$. $Y \in \mathbb{R}^{n \times p}$ is $(\varepsilon_1, \varepsilon_2)$-approximately 2-critical (AC) for $(BM_{ls})$ if*

$$\|S(YY^T)Y\| \leq \varepsilon_1, \tag{9a}$$

$$S(YY^T) \bullet UU^T + 4\|\mathcal{A}(UY^T)\|^2 \geq -\varepsilon_2 \quad (\forall U \in \mathbb{R}^{n \times p} : \|U\| = 1) \tag{9b}$$

We are ready to formalize the concept of spurious critical points.

**Definition 10.** Let $R_Y \in \mathbb{R}_+$ be fixed and let $Y$ with $\|Y\| \leq R_Y$. Given $\boldsymbol{\varepsilon} = (\varepsilon_0, \varepsilon_1, \varepsilon_2) \in \mathbb{R}^3_+$, $Y$ is *spurious* $\boldsymbol{\varepsilon}$-AC if $Y$ is $(\varepsilon_1, \varepsilon_2)$-AC for $(BM_{ls})$ but $YY^T$ is not $\boldsymbol{\varepsilon}'$-approx. optimal for $(SDP_{ls})$ for $\boldsymbol{\varepsilon}' := (\varepsilon_0, R_Y\varepsilon_1, 5\varepsilon_2)$. $Y$ is spurious *exactly* critical if the above holds for $\boldsymbol{\varepsilon} = 0$.

We assume here that $b \in \mathbb{R}^m$ is fixed, and we vary the linear map $\mathcal{A}$. We identify $\mathcal{A}$ with the tuple of matrices $(A_1, \ldots, A_m)$, and hence view it as an element of the Euclidean space $(\mathbb{S}^n)^m$. We assume that $\mathcal{A}$ is *uniformly* distributed on the Euclidean ball $\mathbf{B}_\sigma(\bar{\mathcal{A}}) \subset (\mathbb{S}^n)^m$ of radius $\sigma$ centered at $\bar{\mathcal{A}}$. Consider the set $\mathscr{A}_\varepsilon \subset (\mathbb{S}^n)^m$, consisting of all $\mathcal{A}$ for which there is a spurious AC point:

$$\mathscr{A}_\varepsilon := \{\mathcal{A} \in (\mathbb{S}^n)^m : \exists Y \text{ a spurious } \varepsilon\text{-AC point}\}.$$

We show that if $\tau(p) > m$, then the probability $\Pr[\mathcal{A} \in \mathscr{A}_\varepsilon] \to 0$ as the ratio $\varepsilon_1/\sqrt{\varepsilon_2} \to 0$.

**Theorem 7** (critical $\Rightarrow$ feasible). *Let $p$ such that $\tau(p) > m$, and $\varepsilon \in \mathbb{R}^3_+$. Let $\mathcal{A}$ be uniformly distributed on the Euclidean ball $\mathbf{B}_\sigma(\bar{\mathcal{A}})$. Then*

$$\Pr[\mathcal{A} \in \mathscr{A}_\varepsilon] \leq 4e\, \delta^{\tau(p)-m}\, (3\kappa)^m\, (2n^3 m/\sigma\varepsilon_0)^{\tau(p)},$$

*where $\delta := \varepsilon_1 R_A/\sqrt{\varepsilon_2}$, $\kappa := 2(R_A R_Y^2 + \|b\|) R_A$, and $R_A := \|\bar{\mathcal{A}}\| + \sigma$, provided that $\delta < \sigma\varepsilon_0/4n^3 m$.*

As before, we can derive a high probability bound when $\tau(p) > (1+\eta)m$.

**Corollary 2.** *Consider the setup from Theorem 7. Assume that $\tau(p) \geq (1+\eta)m + \eta t$, $\varepsilon_0 \geq \rho\sigma$ and $\delta \leq (1/3\kappa)^{1/\eta}(\rho\sigma^2/2n^3 m)^{1+1/\eta}$ for some $\eta, t, \rho > 0$. Then $\Pr[\mathcal{A} \in \mathscr{A}_\varepsilon] \leq 4e\,(\rho\sigma^2/6\kappa n^3 m)^t$.*

As in Section 3, our proof of Theorem 5 relies on a geometric characterization of the spurious AC points. First consider the simpler case of spurious *exactly* critical points ($\varepsilon = 0$). The following equation is a consequence of our analysis:

$$\exists \text{ (spurious exactly critical point)} \quad \Longrightarrow \quad \mathbb{S}^n_{n-p} \cap \operatorname{Im}\mathcal{A}^* \text{ is nontrivial.}$$

This implies that the set $\mathscr{A}_0$ has measure zero when $\tau(p) > m$. In order to handle the case $\varepsilon > 0$ we rely again on tubular neighborhoods, as stated next.

**Proposition 2.** *Let $\delta := \varepsilon_1 R_A/\sqrt{\varepsilon_2}$, $D_\lambda := \{\lambda \in \mathbb{R}^m : 2\varepsilon_0 \leq \|\lambda\| \leq R_\lambda\}$, $R_\lambda := 2(R_A R_Y^2 + \|b\|)$. Then for $\mathcal{A} \in \mathscr{A}_\varepsilon$ we have that $\operatorname{tube}_\delta(\mathbb{S}^n_{n-p}) \cap \mathcal{A}^*(D_\lambda) \neq \emptyset$.*

## 5 Polynomial time guarantees

We are ready to derive polynomial time guarantees for the Burer-Monteiro method. The proof of Theorem 1 relies on two facts: (i) the method from Theorem 4 finds an AFAC point of (*BM*) in polynomial time; (ii) Corollary 1 shows that an AFAC point of (*BM*) is approximately optimal for (*SDP*) with high probability. Hence, the Burer-Monteiro method solves (*SDP*) in polynomial time in the setting of smooth analysis. The proof of Theorem 2 is analogous, except that it relies on Theorem 3 and corollary 2.

Appendix C provides explicit complexity estimates for solving (*SDP*) and (*SDP$_{ls}$*) using the Burer-Monteiro method, indicating the precise optimality guarantees of the output.

## 6 Experiments

We present some experimental results to complement our theorems. We rely on the library NLopt [24] (MIT license) to solve the nonlinear problem (*BM*). Concretely, we use the *augmented Lagrangian method* (ALM) implemented in NLopt (which is based on [8]), and we use the preconditioned truncated Newton method as the subroutine. We also rely on the commercial solver Mosek for SDPs.

For our first experiment we consider a random SDP with a planted solution. More precisely, we take a matrix $X_0 \in \mathbb{S}^{50}$, $X_0 \succeq 0$ of rank $r \in \{4, 7, 12\}$, and generate a random SDP for which $X_0$ is an optimal solution. To do so, we generate $m := \tau(r)$ random constraints that are satisfied at $X_0$, and then find a cost matrix $C$ in the normal cone of $X_0$. We generate 100 random SDPs for each $r$. We solve these SDPs with the Burer-Monteiro method, using different values of $p$ (the rank of $Y$) and random initializations. The initial points are matrices with i.i.d. normalized Gaussian entries. Figure 1 shows the percentage of experiments solved correctly for each value of $r$ and $p$. We regard an experiment as "correct" if the criticality conditions from (4) are satisfied.

Figure 1 illustrates that there is a sharp phase transition at the Barvinok-Pataki bound $p = r$. Above the Barvinok-Pataki bound, the Burer-Monteiro method solves most instances. Beneath the Barvinok-Pataki bound, it is not just that our techniques stop working, but that the method itself usually

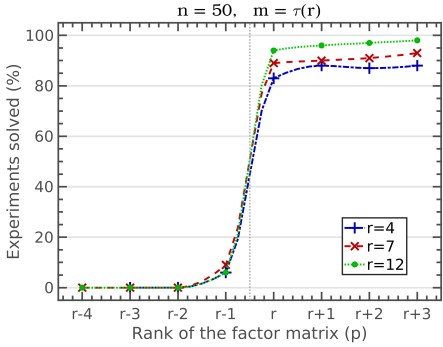

Figure 1: Performance of the BM method for random SDPs with a planted solution of rank $r$.

fails. Note that previous work [5, 30] required $p$ to be larger than $3r/\sqrt{2}$. However, we see from experiments that the phase transition is much sharper than this. Observe that, even for $p \geq r$, the number of experiments solved is always below $100\%$. Nonetheless, the number of bad instances seems to get smaller for larger values of $p$. This agrees with our result from Theorem 5.

For the second experiment we fix the parameters $n := 50$, $m := 28$, and $p := r := 7$. Among the 100 random SDPs considered in Figure 1, we take an instance for which the Burer-Monteiro problem performed badly. We then perturb this seemingly bad SDP by adding varying amounts of noise $\sigma$. For each noise level we solve 70 random experiments, in which both the perturbations and the initializations are random. The perturbations consist in adding to each matrix a random matrix with i.i.d. Gaussian entries scaled by the noise level. Figure 2 summarizes the results obtained.

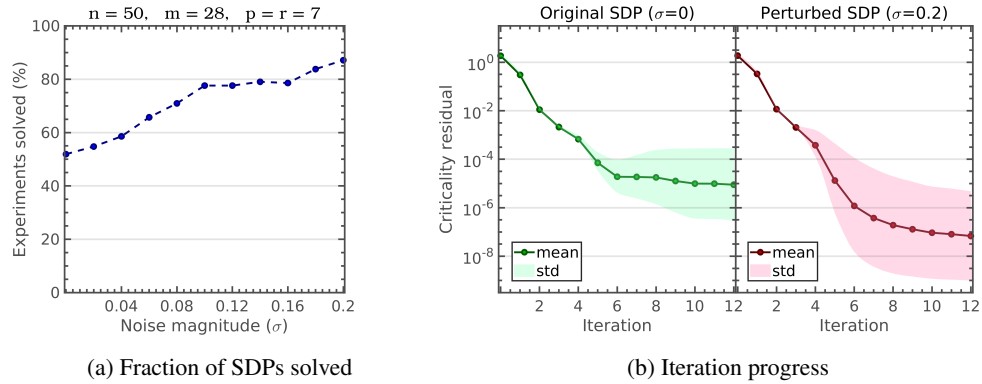

(a) Fraction of SDPs solved             (b) Iteration progress

Figure 2: Performance of the BM method after perturbing a "bad" SDP with different levels of noise.

Figure 2a shows the percentage of instances for which the Burer-Monteiro method succeeded for each noise level. The percentage is with respect to the random perturbation and the random initialization. For the unperturbed problem ($\sigma = 0$) the method succeeds only for $52\%$ of the random initializations. This percentage increases as we add noise. For $\sigma = 0.2$ the method succeeds $87\%$ of the time. Figure 2b shows the progress of the algorithm for the cases $\sigma = 0$ and $\sigma = 0.2$. The progress is measured in terms of the residual of the criticality conditions. The figure shows the mean and standard deviation of the residual for each iteration of ALM.

Figures 2a and 2b illustrate the advantages of smoothing a badly behaved SDP. Theorem 9 predicts that the complexity of the algorithm is proportional to $\sigma^{-d}$ for some exponent $d$. So for a fixed number of iterations $N$, we should set the noise level proportional to $N^{-1/d}$. However, our bounds were shown for the algorithm from Theorem 4. We do not know if they also apply to ALM.

## 7 Discussion

We study the Burer-Monteiro method for solving the problems (*SDP*) and (*SDP$_{ls}$*). We provide the first polynomial time guarantees, using a smoothed analysis setting. Previous analyses for (*SDP*) did not provide end-to-end guarantees and made structural assumptions on the problem instance, while previous work for (*SDP$_{ls}$*) relied on the RIP condition, a strong assumption.

The main technical novelty of this paper is the use of volumes of tubular neighborhoods of varieties for deriving concentration bounds. This allows our results to hold even as $p$ approaches the Barvinok-Pataki bound arbitrarily close. Although the Barvinok-Pataki bound is optimal for general SDPs, there are better bounds for some special classes of SDPs. An important open question is whether our results could be extended to allow for smaller values of $p$ in those special classes.

The Burer-Monteiro method is not a single algorithm, but a family of algorithms, one for each local optimization method for solving (*BM*). Similarly, our guarantees in Theorems 1 and 2 apply for any local method with 2nd order guarantees. The only method we are aware that provably computes AFAC points of (*BM*) efficiently is the one in Theorem 4. It would be interesting to extend our results to classical optimization methods, such as ALM, which have robust implementations.

## Acknowledgments and Funding Disclosure

Ankur Moitra was supported in part by a Microsoft Trustworthy AI Grant, NSF CAREER Award CCF-1453261, NSF Large CCF-1565235, a David and Lucile Packard Fellowship, an Alfred P. Sloan Fellowship and an ONR Young Investigator Award.

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
