# OpenReview forum: "Polynomial time guarantees for the Burer-Monteiro method"
_NeurIPS.cc/2022/Conference — NeurIPS 2022 Accept_

### Official Review · Reviewer_EwVj · 2022-06-29

**Rating:** 8
**Confidence:** 4
**Soundness:** 4 excellent
**Presentation:** 4 excellent
**Contribution:** 3 good

**Summary:**

**Background for the paper**

Consider a general semidefinite program,
\begin{align*}
\min_{X \in \mathcal{C}}C\bullet X,
\end{align*} where $\mathcal{C} = {X\succeq 0, \mathcal{A}(X) - b \in 0^{m_1} \times \mathbb{R}_+^{m_2}}$.

In this summary, we shall refer to this problem as (SDP). The linear operator
$\mathcal{A}(X):\mathbb{R}^{n\times n}\rightarrow\mathbb{R}^{m}$ (where $m = m_1 + m_2$ is the total number of linear constraints) has as its $i$'th element,
$A_{i}\bullet X.$ SDPs form a popular class of convex programs but suffer from large input size in most useful settings. To circumvent this issue of size, Burer and Monteiro proposed expressing the problem variable as
$X=YY^{\top},$ for some $Y\in\mathbb{R}^{n\times p}$ and, instead of (SDP), solving the following proxy problem, which we refer to as (BM):
\begin{align*}
\min_{Y\in\mathbb{R}^{n\times p}, YY^\top \in\mathcal{C}}C\bullet YY^{\top}.
\end{align*}

The benefit conferred by
this reduction is that (BM) uses $O(np)$ memory, whereas (SDP) uses $O(n^2)$.

(BM) clearly lies in a lower rank space than (SDP), and therefore it shouldn't necessarily be the case that solving (BM) solves (SDP). However, it turns out that when $p$ (in the size of $Y$ above) satisfies $p  = O(\sqrt{m})$, then a solution to (BM) also gives one for (SDP). This follows from the celebrated result of Barvinok and Pataki (independently obtained) that there
exists a rank-$\sqrt{m}$ solution to (SDP).

**What problem this paper studies (and solves)**

The tradeoff to the above-mentioned memory saving, though, is that (BM) is a non-convex
problem. Therefore, standard convex optimization algorithms do not give any provable theoretical guarantees for obtaining a global minimum of (BM). This is at odds with the tremendous empirical success that (BM) has found in real-world SDPs, and much effort has been expended on explaining this success.

This paper **gives the first polynomial
time guarantee (in the smoothed analysis setting)  that solving (BM) also solves (SDP)**. Here, by "solving (BM)", we mean finding a second-order critical point.

Previous works include the following. Boumal, Voroninski, and Bandeira
proved that *asymptotically*, (BM) has no spurious 2-critical points under certain technical
assumptions on the problem and when $\tau(p)\geq m.$ However, practical
optimization methods do not attain exact 2-criticality. Pumir, Jelassi,
and Boumal gave guarantees with approximate criticality, with worse
bounds on $\tau(p)$ than guaranteed by Barvinok-Pataki. Finally, there's the work of Bhojanapalli, Boumal, Jain, Netrapalli, that provides similar guarantees for the penalized (SDP).

This paper's specific contribution is to **provide a polynomial-time guarantee that solving (BM) (under the Barvinok-Pataki limit) provides a solution to (SDP)**.

**Questions:**

1. My main question to the authors is to please clarify how the paper compares against the result of Bhojanapalli, Boumal, Jain, and Netrapalli mentioned in the paper. Thank you.

**Strengths And Weaknesses:**

**ORIGINALITY**

I believe this paper scores highly on originality. At the heart of their result are the following two sub-results.

1. Theorem 4, which gives a polynomial-time bound on computing 2-AFAC points. The algorithm this theorem is based on first finds an approximately feasible point, then iteratively minimizes a Lagrangian-style function and updates its guess of the minimum.
2. Theorem 5, which proves that a 2-AFAC point for (BM) is approximately optimal for (SDP). To do so, the paper characterizes the set of spurious AFAC points (i.e., the set of symmetric $C$ such that the tuple $(Y, \lambda)$ is a spurious AFAC pair) geometrically. A characterization of the simpler set of EFAC points may be done via simply the KKT conditions and simple arguments about appropriate subspaces. However, extending this to AFAC points is highly non-trivial and requires technical machinery such as computing the volume of tubular neighbourhoods around certain varieties.

**QUALITY**

I think, while the contributions are significant, the paper can improve its quality of related literature review. Given that a lot of the work done on this problem involves subtle differences of definition of criticality, I think that a critical piece of content missing from this paper's current version is a detailed comparison against some other (already published) works.

For example, I think the paper must clearly explain why the work of Bhojanapalli, Boumal, Jain, and Netrapalli doesn't already give the stated result (albeit using a different technique; I agree that this paper's technique is novel). In particular, maybe the paper should elaborate on lines 99-102 a bit more: how is solving the penalized version that different from solving the paper's version? I'll be happy to increase my score on this aspect (and, thus, the overall score) if this were explained well. (I agree that BBJN may not have explicitly provided a final runtime but it's perhaps still possible to trivially deduce it from there; so I think it's important to understand why this isn't the case).

**CLARITY**

The paper is generally quite clearly written and well motivated.

**SIGNIFICANCE**

The work is of great significance. SDPs are useful in a plethora of fields, and (BM) is a standard technique used to solve them in practice. A theoretical understanding of *why* (BM) works despite the inherent non-convexity could not only inspire faster algorithms for SDPs, but also provide a deeper understanding of non-convex algorithms in general.

---

> ### Author Response · Authors · 2022-08-02
> **Reply to Reviewer EwVj**
>
> Thank you very much for your feedback. We answer your question below. Please ask follow-up questions if anything is unclear.
>
> * Explain why the work in [5] doesn't already give the stated result. How is solving the penalized version that different from solving the paper's version? (BBJN may not have explicitly provided a final runtime but it's perhaps still possible to trivially deduce it).
>
> The paper [5] does not provide end-to-end polynomial time guarantees for solving SDPs because of a few issues. First of all, the method in [5] requires a preprocessing step that in general is as hard as solving another SDP, see the paragraph in [5, pg 9] after equation (13). Since solving an auxiliary SDP is required, [5] fails to explain why the BM method improves over general SDP solvers.
>
> Secondly, the complexity analysis is nontrivial because the penalized function has high condition number, which actually depends on the inverse of the desired tolerance $\epsilon$ (this will be discussed below). Consequently, standard complexity estimates like Theorem 3 from our paper do not apply.
>
> The last issue, which is more technical, is that [5, Thm 13] shows that the obtained matrix $UU'$ is approximately optimal for the penalized problem, but instead we would like to show that $UU'$ is approximately optimal for the constrained problem. In particular, we need to bound the infeasibility error of $UU'$. This requires tuning the penalty parameter $\mu$ based on the desired tolerance $\epsilon$. However, $\mu$ is also constrained by the assumptions of [5, Thm 13] (in particular, $\mu \epsilon^{3/2} \leq O(\gamma k^2 \sigma^2)$). Satisfying all requirements is tricky, and the straightforward analysis leads to results that are only valid when the perturbation magnitude $\sigma$ is large (while in smoothed analysis we want small perturbations).
>
> For clarity, we provide a simple approach toward bounding the infeasibility error from [5, Thm 13]. Let $UU'$ be the obtained matrix, $X$ be the optimal solution of the penalized problem, and let $r(UU') = \|A(UU')-b\|$, $r(X) = \|A(X) - b\|$ be their feasibility residuals. By [5, Thm 13], we have that
>     $<C, UU'-X> + \mu (r(UU')^2 - r(X)^2) \leq \gamma \sqrt\epsilon Tr(X) + .5 \epsilon \|U\|.$
> Then to bound the infeasibility error $r(UU')$ by $\epsilon$ one needs to set
>     $\mu = \epsilon^{-2}(<C,UU'-X> + \gamma \sqrt\epsilon Tr(X) + .5 \epsilon \|U\| + \mu r(X)^2).$
> For simplicity, let us say that $\mu = \Omega(\epsilon^{-2})$ (this requires justification). Taking into account the constraint $\mu \epsilon^{3/2} \leq O(\gamma k^2 \sigma^2)$, leads to $\sigma^2 = \Omega(\epsilon^{-1/2}/\gamma)$, so the perturbation magnitude is large.
>
> To sum up, it is possible that the work of [5] can be extended in order to provide end-to-end guarantees for solving SDPs. However, this would require fixing multiple issues: the preprocessing, the missing complexity analysis, and the constrained optimality guarantees. This would be a nontrivial extension of their results.
>
> We will further explain the differences with [5] in the final version of the paper. Thanks again for the question.

---

> > ### Comment · Reviewer_EwVj · 2022-08-08
> > **Thank you for clarifying the difference from BBJN. Additional question (if possible to answer)**
> >
> > ### Thank you for the response
> > I am grateful to the authors for the very detailed response. So basically the problem is that translating BBJN's solution, which is a solution of the penalty formulation, to one that's a solution for the constrained case (i.e., exact feasibility) requires tuning the parameter that scales the penalty in a way that we get desired feasibility. Working out these calculations gives a very large perturbation. (I hope I got that right).
> >
> > ### Suggestion to add the explanation you gave about BBJN in your manuscript for better clarity
> > I think it's still worthwhile trying to explain the difference explicitly in the paper since it's not obvious that the two cases are so different, especially given that in optimization algorithms, we switch back and forth between constrained and unconstrained (via Lagrange multipliers) problem formulations.
> >
> > ### Increasing my score
> > In any case, thank you again! I'm raising my score to an 8 based on your explanation.
> >
> > ### Final question (if possible)
> > I apologize for my delay in responding, so I won't penalize the authors if they don't get to the following question, but if they are able to, then I'm curious: "For the runtime of BM, what do you think is the right dependence on the perturbation? Currently this is polynomial in the inverse perturbation, do you think this can be made polylogarithmic?"

---

> > > ### Author Response · Authors · 2022-08-08
> > > **Answer to additional question**
> > >
> > > Thanks again for your feedback on the paper. We will follow your suggestion of expanding on the difference between our paper and BBJN, particularly regarding the distinction between the unconstrained and constrained formulation.
> > >
> > > Regarding the dependence on the perturbation, in smoothed analysis we usually get inverse polynomial dependence on the perturbation. In some cases, like linear programming, algorithms are based on solving the problem to high accuracy and being able to "jump" from there to an optimal solution (e.g., see the book of Grotschel, Lovasz and Schrijver). So it would actually be quite surprising if you could get the dependence on the perturbation to be poly logarithmic, because it would mean you can take the perturbation to be so small that when you solve the perturbed problem, you can jump to the solution of the unperturbed problem. And then you wouldn't really need the perturbation in the first place. In contrast, smoothed analysis is often used to get around worst case lower bounds, like with the simplex method. The same is true in our setting since the Beurer-Monteiro method was recently shown to fail beneath the Barvinok-Pataki bound in general. Thus smoothing, or some other way to get around worst case analysis, really is necessary. Hence, we believe that the polynomial dependence on the perturbation is expected.

---

### Official Review · Reviewer_yGpw · 2022-07-01

**Rating:** 7
**Confidence:** 2
**Soundness:** 3 good
**Presentation:** 3 good
**Contribution:** 3 good

**Summary:**

The authors propose the Burer Monteiro method to solve an SDP problem in polynomial time up to any degree of accuracy. In the Burer Monteiro method, the idea is to write a symmetric matrix as $YY^{T}$ which then leads to a non-convex program. The primary technique is by slight perturbation of the cost matrix and the constraints so that they avoid a measure zero "bad set". The authors show that under certain assumptions, spurious approximately critical points can be avoided by using sophisticated techniques from real algebraic geometry.

**Questions:**

1) What is the main advantage that is obtained by writing the SDP in form of BM? Also, how is the BM_{ls} problem solved in order to get a feasible solution Y? It would be good to write down the exact optimization algorithms for solving BM and BM_{ls}.

2) Why is it necessary to go through this route? Can the same optimization techniques for solving BM_{ls} and then BM be used to solve SDP_{ls} and then SDP directly?

**Limitations:**

There are no apparent limitations

**Strengths And Weaknesses:**

The paper is well-written. Unfortunately, I am not very well-versed in this area to point out weaknesses.

---

> ### Author Response · Authors · 2022-08-02
> **Reply to Reviewer yGpw**
>
> Thank you very much for your feedback. We answer your questions below. Please ask follow-up questions if anything is unclear.
>
>
> * What is the main advantage that is obtained by writing the SDP in form of BM? How is BM_{ls} solved to get a feasible solution Y? It would be good to write down the algorithms for solving BM and BM_{ls}.
>
> The main advantage of writing the (SDP) in the form of (BM) is that it requires significantly less memory (n x p entries, instead of n^2). Regarding the algorithms, our paper attempts to be algorithm agnostic as much as possible. Our results apply as long as (BM) and (BM_ls) are solved using algorithms that satisfy certain polynomial time guarantees. There are a few possibilities for solving (BM_ls), e.g., the ARC method (Thm 3). For problem (BM), we are only aware of Algorithm 1 from Appendix A.
>
>
> * Can the same optimization techniques for solving BM_{ls} and then BM be used to solve SDP_{ls} and then SDP directly?
>
> The problem of solving the SDP directly is that the matrix X has too many entries to store. Factorizing the matrix as X = YY' is a simple way to reduce the memory requirements. There exist alternative ways to improve memory, but the BM method is one of the simplest and most widely used techniques.

---

> > ### Author Response · Authors · 2022-08-08
> > **Any further clarification?**
> >
> > Please let us know if you still have any doubts about our paper, so we can clarify them before the end of the discussion period.

---

### Official Review · Reviewer_1Yjq · 2022-07-10

**Rating:** 7
**Confidence:** 2
**Soundness:** 3 good
**Presentation:** 4 excellent
**Contribution:** 4 excellent

**Summary:**

This work studies the complexity of the Burer-Monteiro method for solving semidefinite programs (SDP). They show that in the smoothed analysis setting (initiated by the seminal work of Spielman and Teng), under some assumptions (certain parameter regimes, compactness, etc), the algorithm terminate in polynomial time. In more details, the Burer-Monteiro method is a nonconvex practical method for solving SDPs, that is useful in applications. For SDP, it can be thought of as trying to directly compute the Cholesky decomposition X = YY^T of the desired PSD matrix variable X. The Barvinok-Pataki bound gives a bound on the dimensions of Y (can be thought of as the rank) above which this approach should work. Prior work has considered various settings and studied when optimization methods based on this approach will succeed and not get stuck on local optima. Also, the smoothed analysis setting has been considered before (although with stronger bounds on the rank) and termination guarantees were obtained. Other works have obtained polynomial runtime guarantees, under various assumptions such as the restricted isometry property. In this work, they obtain polynomial time guarantees for the B-M method under various analytic assumptions and in the smoothed-analysis setting, getting almost close to the Barvinok-Pataki bound.

The full method analyzed here has two stages, first to get a feasible point and then use it as initialization to get an optimal point. In the first stage, under random perturbations on the constraint matrices and other analytic assumptions, 2nd-order methods are shown to terminate in polynomial time (theorem 2). In the second stage, under random perturbations to the cost-matrix and various analytic assumptions, it's shown that 2nd-order methods terminate in polynomial time (theorem 1). Together, this shows polynomial complexity of the B-M method in the smoothed analysis setting, with explicit bounds derived in Appendix C. Both these assumptions crucially assume that the rank is above the Barvinok-Pataki bound. The proofs contain a few parts, including a robust method (modified from prior works) to compute critical points, and arguing that spurious critical points do not exist in the smoothed analysis setting when the random permutation is uniform over a ball. For the latter, the analysis involves integrals over tubes, which has been estimated in prior works. Finally, simple experiments were conducted showing how the B-M method performs around the Barvinok-Pataki bound. And in cases when the algorithm failed, further experiments show random perturbations help, giving support to the theory developed here.

**Questions:**

Some questions were raised above.
- How important is it that the perturbations are uniform? What happens if some other perturbation distribution is used?
- Correctness of algorithms are measured based on criticality conditions (L301). It should be clarified why this is reasonable
- In L584, the sentence seems incomplete

**Limitations:**

Limitations of the theoretical results are implicit in the assumptions made. Negative societal aspects are not raised because since this is primarily a theoretical paper.

**Strengths And Weaknesses:**

Strengths:

This is a very well-written and interesting work that basically offers good evidence why the Burer-Monteiro method works well. Related work is cited well and proper acknowledgement is given where necessary (e.g. L481, L511). To the best of my knowledge, this work is the first to obtain fully polynomial runtime guarantees close to the Barvinok-Pataki bound. Like the authors clarified, other works gloss over certain details such as initialization. The main technical tool is utilizing known bounds on integrals over tubes, while prior works have used simple linear algebra or random matrices. This technical idea is simple but seems new in this setting.

Weaknesses:

The main limitation is that the experiments conducted in this work are not very convincing but this is offset by the fact that the primary contribution of this work is theoretical. In other words, the experiments ofer some evidence to support the theory but are not necessarily very exhaustive. For example, the SDPs are generated randomly satisfying the feasibility of a chosen point. Is it possible that this randomness implicitly helps the B-M method?

---

> ### Author Response · Authors · 2022-08-02
> **Reply to Reviewer 1Yjq**
>
> Thank you very much for your feedback. We answer your questions below. Please ask follow-up questions if anything is unclear.
>
> * The SDPs are generated randomly. Is it possible that this randomness implicitly helps the B-M method?
>
> It is possible that the randomness improves the convergence of the BM method. Nonetheless, the purpose of the experiments is not to test the effectiveness of the method, but rather to illustrate relevant qualitative features of it (namely, the sharp threshold with the rank, and the effect of the smoothed analysis). The experiments illustrate these features, complementing our theoretical results.
>
>
> * How important is it that the perturbations are uniform?
>
> While our analysis is for uniform perturbations, it is possible to derive similar results for Gaussian and other distributions through the use of anti-concentration inequalities. However, the required rank will be higher for other distributions since the real algebraic geometry tools are optimized for the uniform distribution. We point out that the purpose of smooth analysis is to illustrate that the method works well for "most instances", so the precise choice of distribution is not as relevant.
>
>
> * Correctness of algorithms are measured based on criticality conditions. Clarify why this is reasonable
>
> Measuring the quality of a point based on the criticality conditions is a standard practice in optimization methods, particularly those relying on primal-dual updates. In particular, several SDP solvers use criticality conditions to decide when to terminate.
>
>
> * In L584, the sentence seems incomplete
>
> Thanks for noticing this typo. It will be fixed for the final version.

---

### Official Review · Reviewer_Vu42 · 2022-07-11

**Rating:** 7
**Confidence:** 3
**Soundness:** 4 excellent
**Presentation:** 4 excellent
**Contribution:** 3 good

**Summary:**

This paper studies optimization of a general class of SDPs having a linear objective and general linear constraints, using the nonconvex Burer-Monteiro method. It is known that for the BM formulation in dimension n x p, if the rank p and the number of constraints m satisfy the Barvinok-Pataki bound p(p+1)/2 > m, then for generic cost matrices the BM optimization has no spurious second-order critical points, and its global minimum also minimizes the original SDP. The main result of this paper is a more quantitative statement on polynomial-time convergence of optimization algorithms in a smoothed analysis framework, showing that if the cost and constraint matrices are uniformly distributed over balls of radius sigma, and if p(p+1)/2 > (1+eta) m for any fixed constant eta > 0, then an optimization algorithm for the BM problem succeeds in finding an approximate minimizer of the SDP in poly(n, sigma^{-1}) time.

This is shown by characterizing the set of cost matrices for which the BM problem has a near second-order-critical point that is not near-optimal for the SDP, and showing that this set belongs to a thin tube around the measure-zero variety that was previously identified as potentially leading to spurious local optimizers of the BM objective. The volume of this tube is bounded using generic results from algebraic geometry, leading to a statement that the smoothed cost matrix w.h.p. does not admit such a spurious near-critical-point. On this high-probability event, the author(s) show that a generic optimization algorithm can compute a near second-order-critical point in polynomial time, which must then be near-optimal. The algorithm requires a p.s.d. initialization that nearly satisfies the equality constraints, which in general may be difficult to obtain. Thus the author(s) also analyze the optimization of a second unconstrained least-squares SDP using the BM approach, which may be used to obtain such an initialization. This second analysis smooths instead over the constraint matrices, and uses a similar algebraic-geometric proof technique.

**Questions:**

(1) The statements of Theorems 1 and 2 feel a bit too informal and imprecise. It might be clearer to either make them more precise, or to explicitly qualify them as informal versions and provide references to their more formal statements in the supplement.

(2) In the end-to-end guarantee, SDP_{ls} needs to be solved to near-optimality such that Y falls into the set M_beta where the LICQ bound holds, but (unless I missed it) the paper doesn't seem to quantify how small beta needs to be. Is it possible to make a statement about this---e.g. can beta^{-1} be shown to be poly(n, sigma^{-1}), so that it indeed suffices to run the optimization of SDP_{ls} for polynomial time?

More minor clarifications:

Do the authors mean to write the second equality in Proposition 1? I don't see why this is an equality, and also it doesn't seem to be used in the subsequent proof.

display after l582, the second = should also be >=

display after l660, what are eps_0', eps_1', and eps_2' here?

**Limitations:**

Yes

**Strengths And Weaknesses:**

I think the paper makes a nice contribution to the literature. The main results are perhaps not too surprising, but they do seem to take some work and insight to obtain, especially down to the sharp Barvinok-Pataki threshold. Defining the right notions of "near-criticality" for these SDPs in a way that connects both to the volume-of-tubes arguments being used and to rigorous polynomial-time guarantees for optimization algorithms seems a bit subtle, and for example required the development and analysis of a new variant of a generic second-order optimizer with equality constraints (Theorem 4). The proofs are clean, the paper resolves a very natural question, and I think it's above the bar for NeurIPS.

---

> ### Author Response · Authors · 2022-08-02
> **Reply to Reviewer Vu42**
>
> Thank you very much for your feedback. We answer your questions below. Please ask follow-up questions if anything is unclear.
>
> * It might be clearer to either make Theorems 1 and 2 more precise, or to explicitly qualify them as informal versions and provide references to their more formal statements in the supplement.
>
> We will follow the second approach, including a reference to the full theorems (including explicit constants) in the appendix.
>
>
> * The paper doesn't seem to quantify how small beta needs to be. Is it possible to make a statement about this?
>
> Our results make two assumptions on the feasible set, namely, that it is compact and smooth (satisfies LICQ). Moreover, we need the compactness and smoothness constants to be polynomially bounded (e.g., to avoid exponential size solutions). The required value of \beta is determined from the smoothness constant. More precisely, \beta is approximately R /(sum_i |A_i|), where R is the smallest LICQ constant among all feasible points.
>
> As the reviewer points out, the current version of the paper does not discuss how pick \beta, but we will discuss it for the final version of the manuscript.
>
>
> * Do the authors mean to write the second equality in Proposition 1? It doesn't seem to be used in the subsequent proof.
>
> The second equality is correct, but it is indeed not really needed. We will remove it from the paper.
>
>
> * After l582, the second = should also be >=
>
> Thanks for noticing. We will fix it in the final version.
>
>
> * After l660, what are eps_0', eps_1', and eps_2' here?
>
> These are the errors bounds obtained in Theorem 10, but we forgot to point it out. We will clarify this for the final version.

---

> > ### Author Response · Authors · 2022-08-08
> > **Any further clarification?**
> >
> > Please let us know if you still have any doubts about our paper, so we can clarify them before the end of the discussion period.

---

### Meta-Review · Area_Chair_3ecn · 2022-08-26

**Recommendation:** Accept
**Confidence:** Certain

**Metareview:**

This paper gives polynomial time smoothed analysis guarantees for the Burer-Monteiro method. The result is new and interesting. Prior works fall short of this goal for various reasons. Boumal et al. which started this line of work gives non-robust generic guarantees, while Bhojanapalli et al. look at a penalized version that's different (and suffers from ill-conditioning). While the basic approach is to take the arguments in Boumal et al. and make them robust in a natural way, there is effort in formalizing this using volumes of certain tubes and machinery from algebraic geometry. This paper is a solid contribution to the literature on the Burer-Monteiro method.

**Award:**

No

---

### Decision · Program_Chairs · 2022-09-14

Accept